# English version of the Computer Vision Symptom Scale (CVSS17): Translation and Rasch analysis-based cultural adaptation

**Mariano González-Pérez**[1¤*], **Carlos Pérez-Garmendia**[2], **Kathleen Hoang**[3], **Rosario Susi**[4], **Beatriz Antona**[1], **Ana-Rosa Barrio**[1], **Mark Rosenfield**[5]

**1** Optometry and Vision Department, Faculty of Optics and Optometry, Complutense University of Madrid, Madrid, Spain, **2** DXC Technology Company, Madrid, Spain, **3** Downtown Eyecare, New York, United States of America, **4** Department of Statistics and Data Science, Faculty of Statistical Studies, Complutense University of Madrid, Madrid, Spain, **5** SUNY College of Optometry, New York, United States of America

¤ Current address: Faculty of Optics and Optometry, Complutense University of Madrid, Madrid, Madrid, Spain

* marian06@ucm.es

## Abstract

### Background

Because the CVSS17 was originally developed in Spanish, the objective of this study was to adapt it linguistically and culturally into English while evaluating its psychometric properties.

### Methods

After translating and adapting the CVSS17 to English, 441 participants (aged 18 to 65 years) from a general population, recruited from an on-line panel, completed the English version (CVSS17$_{ENG}$). To determine the measurement properties of CVSS17$_{ENG}$, we used the partial credit model. To assess convergent validity, coefficients of correlation between CVSS17$_{ENG}$ and the Ocular Comfort Index or Visual Discomfort Scale were calculated. A subset of 218 subjects was tested for test-retest reliability. In addition, differences between CVSS17$_{ENG}$ and CVSS17 were tested through Differential Item Functioning (a Rasch statistic used to check item bias).

### Results

A total of 441 responses to CVSS17$_{ENG}$ (average age, 38.57; age range, 19–65; females, 50.24%) showed good fit to the Rasch model, good precision (person separation index = 2.73), and suboptimal targeting (-1.43). Residual principal component analysis suggested multidimensionality, but this was ruled out by a disattenuated correlation coefficient value of 0.82, and no DIF according to sex or age was found. Pearson correlation for CVSS17$_{ENG}$-VDS was 0.54 (p < 0.01) and Spearman correlation for CVSS17$_{ENG}$-Ocular Comfort Index was 0.66 (p < 0.001). For test–retest reliability, the limits of agreement were 9.39 and -8.61. Rasch analysis results were similar for CVSS17 and CVSS17$_{ENG}$ and there was no evidence of item bias based on gender or age.

**Data availability statement:** Data is available at https://doi.org/10.6084/m9.figshare.25909171.v1.

**Funding:** BA, AB, MG-P and RS received a grant from the Instituto de Salud Carlos III through Project PI18/00374 (co-funded by European Regional Development Fund "A way to make Europe"). DXC Technology Company, Madrid provided support in the form of a salary for CP-G. Downtown Eyecare, New York provided support in the form of a salary for KH. ALAIN AFFLELOU Óptico provided support in the form of a salary for MG-P The specific roles of these authors are articulated in the "author contributions" section. The funders didn't play any role in the study design, data collection and analysis, decision to publish, or preparation of the manuscript.

**Competing interests:** The authors have read the journal's policy and have the following competing interests: Carlos Pérez-Garmendia is an employee of DXC Technology Company, Madrid. Kathleen Hoang is an employee of Downtown Eyecare, New York. Mariano González-Pérez was an employee and received consultancy fees from ALAIN AFFLELOU Óptico during the project. There are no patents, products in development, or marketed products associated with this research to declare. This declaration does not alter the authors' adherence to PLOS ONE policies on sharing data and materials.

## Conclusion

The English version of CVSS17 demonstrates comparable performance to the original, indicating its suitability for clinicians and researchers to reliably assess Digital Eye Strain among English-speaking users of screen-based electronic devices.

## Introduction

Known as Digital Eye Strain(DES) [1], the set of visual and ocular symptoms associated with prolonged use of screen-based digital devices is prevalent in optometric practice and extends beyond the workplace due to the widespread use of digital devices for both social and professional activities [2].

The first Rasch-based patient-reported outcome instrument (PRO instrument) for measuring DES, the Computer Vision Symptom Scale (CVSS17), was published in 2014 [3]. It consists of 17 items designed to gather information on 15 different symptoms. CVSS17 is available in hard copy and online at the CVSS17 website (cvss17.com), where anyone can access the questionnaire, complete it, and instantly obtain their CVSS17 scores, which can be categorized into five levels of severity [4].

CVSS17 has shown significant advantages over existing assessment tools, notably the widely utilized Computer Vision Syndrome Questionnaire (CVS-Q). Researchers in various clinical studies [5–8] have chosen CVSS17 to overcome the limitations identified in CVS-Q. While CVS-Q exhibits suboptimal item-person targeting [9], CVSS17 shows a higher measurement precision [4]. Furthermore, CVSS17 allows for independent scoring of the two specific components of DES [10]: 1) the vision related symptoms, also known as Internal Factor Symptoms [11], and 2) the ocular surface related symptoms, also known as External Factor Symptoms [11]. Moreover, it provides scores that can be grouped into five different levels of severity [4]. These properties establish CVSS17 as an excellent option for comprehensive and detailed assessments in both research and clinical contexts.

As there are almost 600 million native English speakers worldwide (source: https://www.worlddata.info/languages/english.php), and English is the dominant language in scientific publications, an English version of the CVSS17 was necessary as the original was developed in Spanish. An in-house translated English version was provided in 2014 as supporting information. However, a good translation alone is not sufficient to ensure that the translated version preserves the psychometric properties of the original version. Therefore, a cross-cultural adaptation process was required, involving the development of versions of the original questionnaire that are equivalent to the original but are also adapted linguistically and culturally to a different context [12].

The objective of the present study was to adapt the CVSS17 to English and assess the performance of the new version by examining its main psychometric properties through Rasch analysis. For this purpose, an English version ($CVSS17_{ENG}$) was developed in accordance with instructions mentioned in several guidelines [12–14] and previous cross-cultural adaptations [15–17].

## Materials and methods

The original CVSS17 comprises 17 items that query about the frequency, intensity and discomfort associated with 15 different symptoms. A scoring table based on Rasch analysis assigns scores to each item, resulting in a final score ranging from 17 (least symptomatic) to 53 (most symptomatic) [3]. Final scores are categorized into five levels of severity, and the

questionnaire can be subdivided into two sub-scales to independently evaluate either visual or ocular symptoms [4].

To accomplish our objectives, we conducted a two-stage investigation:

1) Stage 1. Translation and transcultural adaptation of CVSS17. The original CVSS17 questionnaire was translated and adapted into English to create the English version of CVSS17 (CVSS17$_{ENG}$). The translated version was pretested on a small sample, assessing the survey questions, and conducting preliminary Rasch analysis.

2) Stage 2. Validity analysis, reliability analysis of the CVSS17$_{ENG}$ and comparison with the original version. In this stage, we administered CVSS17$_{ENG}$ to a larger sample of the USA population and analyzed responses using the Partial Credit Model in Rasch analysis. Spanish and English versions were compared to identify any Differential Item Functioning (DIF).

The study details are summarized in Fig 1.

## Stage 1. Translation and transcultural adaptation of CVSS17

In this stage, it was crucial to ensure cultural equivalence and appropriateness of the translated items. To achieve this, it was necessary to assemble a research team that included members (see supplementary information S1 File) with previous experience in cross-cultural adaptations of heath symptoms questionnaires. The team comprised the study coordinator (MG), clinicians who are native of English or Spanish (KH, MR, MG), professional translators (AB and RS) and other professionals able to provide relevant insights on technical terms (RS). Furthermore, the translation and cross-cultural adaptation process involved five steps [13, 14] which included:

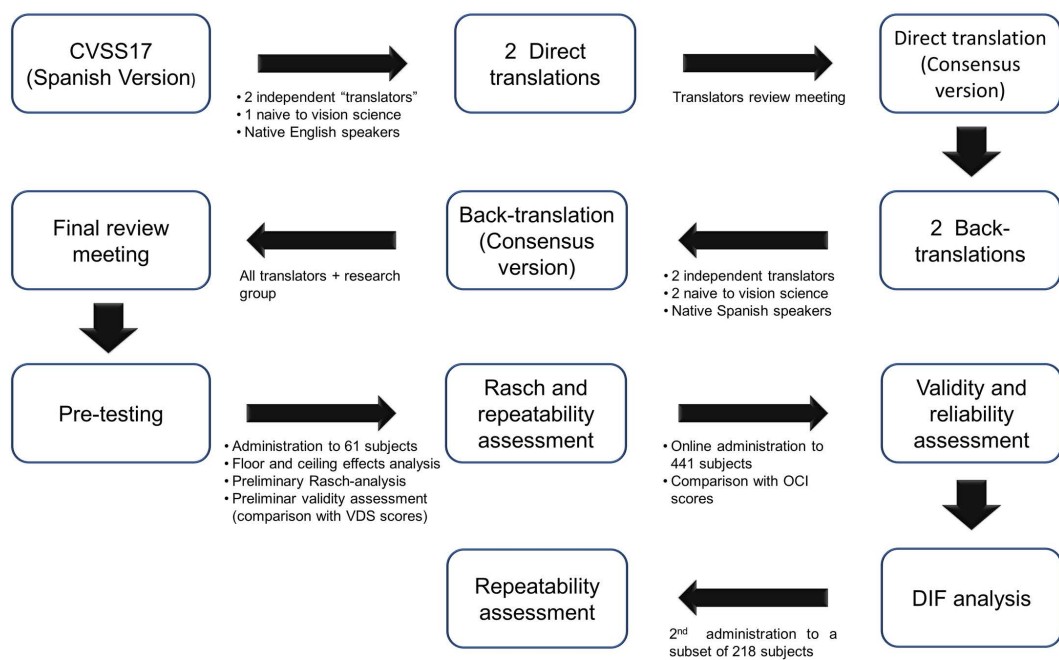

**Fig 1. Flow diagram showing the study details.**

1. Direct translation by two bilingual translators: The original CVSS17 version underwent an independent translation by two bilingual translators: a professional translator and a staff member from the State University of New York (SUNY). These translators were unfamiliar with the explored concepts and had English as their first language.

2. Development of a consensus version based on the two direct translations: The team conducted a meeting to consolidate and merge the two direct translations into a unified forward translation.

3. Back translation by two native Spanish translators: Two additional bilingual translators, whose first language was Spanish and who were blind to the original version, independently translated the consensus version back into Spanish. To avoid information bias, one of the translators were a professional translator naive about the concepts explored.

4. Final review meeting: Translators and research group members convened to integrate the four prior translations, resulting in a pre-final version of CVSS17$_{ENG}$. For CVSS17$_{ENG}$, the panel decided to utilize the Rasch-based scoring table from Spanish CVSS17, that is provided as supporting information (S1 Table).

5. Pretesting of the consensus version: conducted to assess item difficulty in a small sample recruited by convenience from the University staff. A minimum sample size of 50 was chosen to ensure stable item calibrations and person measures within a 1-logit range, as recommended by Linacre [18]. The inclusion criteria were:

- Aged from 18 to 65

- English as the mother tongue

- Use of screen-based electronic devices (SBED) at least four hours a day and/or over 20 hours a week

Exclusion criteria included:

- Prior non-refractive visual surgery

- Active visual or neurologic diseases, medication affecting vision, or any disability hindering questionnaire comprehension.

- Employment or enrollment in optometry and/or vision sciences.

- Unwillingness to participate in the study

Furthermore, the convergence validity of CVSS17$_{ENG}$ was preliminarily assessed in this sample by comparing its responses with those of the Visual Discomfort Scale (VDS) [19], which was administered alongside CVSS17$_{ENG}$ in a random order.

## Stage 2. Validity analysis, repeatability analysis and comparison with the original version

In the second stage, we conducted an online survey using the research platform provided by Prolific Academic Ltd., a platform designed to assist researchers in enlisting participants for their online research [20]. For the recruitment process, we first specified the demographic characteristics of the target participants: age, location, and language proficiency. In addition, we used Prolific's prescreening questions to filter participants based on the specific criteria relevant to the study (e.g., computer and/or digital device use, occupation, health conditions). All the Prolific's users fulfilling our criteria were invited to participate in the study.

We compensated respondents who completed the survey once with 1.60 USD, and those who completed it twice for repeatability assessment received 4.27 USD

In this stage, we administered the CVSS17$_{ENG}$ and specifically selected participants from the USA, fulfilling the same inclusion and exclusion criteria used for the Spanish version development. I

The inclusion criteria for this stage were:

- Age from 18 to 65

- Normal or corrected-to-normal vision

- SBED use for more than 4 hours a day and/or over 20 hours a week

The exclusion criteria for this stage were:

- Employment or enrollment in optometry and/or vision sciences.

- Unwillingness to participate in the study

All these criteria were analyzed as reported by users in the Prolific database.

A total of 5386 Prolific users meeting these criteria were invited, and our study included the first 220 women and 221 men who accepted the invitation, as a minimum of 400 people were required for precise calibration of the CVSS17$_{ENG}$ [18] and 200 subjects in each group (male-female, Spanish-English) are recommended for DIF Analysis [21].

To assess the repeatability and convergent validity of CVSS17$_{ENG}$, two weeks after the first survey, CVSS17$_{ENG}$ was administered a second time along with an online version of the Ocular Comfort Index (OCI) [22] to a subset of 218 subjects selected at random from the first set of participants. Therefore, intraclass correlation coefficient and within-subject standard deviations was calculated. In addition, the Bland-Altman method was used to determine the mean difference and 95% agreement limits between sessions.

Assuming an expected correlation between CVSS17 and OCI above 0.3, with five percent missing response, a minimum sample size of 90 subjects were required for the convergent validity assessment. Following the recommendations of McAlinden et al. [23], and assuming a 10% precision for studying test-repeatability between two CVSS17$_{ENG}$ administrations, a minimum sample size of 192 subjects was required.

This study was a part of a project aimed to develop a new PRO instrument called EDVOS-CAT and the CVSS17 was adapted to English for the validation phase of this instrument. The project protocol (Protocol number: 17.244-E) was approved by the Research Ethics Committee of Hospital Clínico San Carlos (Madrid, Spain). Recruitment for stage 1 began on March 1st, 2018, and ended in March 31st, 2018. For stage 2, it began on May 16th, 2022.

Participants were required to provide electronic informed consent before accessing the questionnaire. This was done by requiring participants to access the study welcome page (https://cvss17.com/english), which clearly communicated the purpose, research objectives, potential risks and discomforts, anticipated benefits, and formal acceptance of informed consent. Participants also had the option to download a copy of the informed consent form and contact information for the research group to address any questions they may have about participating in the study. Participants were required to acknowledge the information provided before accessing the questionnaire.

## Analysis strategy

Descriptive data, repeatability assessment, factor analysis and coefficient of determination ($R^2$) between CVSS17 and CVSS17$_{ENG}$ item measures were conducted using IBM SPSS Statistics package version 27.0 (Statistical Package for Social Sciences).

For the analysis of the psychometric properties of the CVSS17$_{ENG}$ and CVSS17 conducted in this study, we employed a unidimensional Item Response Theory (IRT) model: the Partial Credit Model (PCM), which is an extension of the Rasch model useful for polytomous items. PCM analyses were carried out using WINSTEPS (Version 4.7.0.0, Winsteps, Beaverton, Oregon, USA).

As the items in the CVSS17 are polytomous, it was necessary to select between the Partial Credit Model (PCM) and the Rating Scale Model (RSM) prior to analysis. The PCM is considered less restrictive, although it necessitates a larger sample size and may complicate communication with users, comparison with other similar instruments, and so forth [24]. The CVSS17 comprises items asking about various symptoms with different response options. For instance, some items inquire about the intensity of symptoms, whereas others ask about frequency. Moreover, according to the CVSS17 Rasch-based scoring system, these categories contribute differently to the overall scale score [3]. For instance, category 4 of item A32 adds four points, whereas category 4 of item A30 adds two points. Consequently, given that the RSM model assumes a uniform distribution of response categories across all items, the PCM was chosen to ensure the highest degree of accuracy.

As in previous questionnaire's cross-cultural adaptations [15–17,22], Rasch analysis results were used to determine:

1. Rating scale performance: assessed by examining the order of category thresholds. Disordering happens when response options deviate from the expected hierarchical order.

2. Item fit statistics (Infit and Outfit mean square fit): indicate the degree to which items in a domain align with Rasch model expectations. According to the criteria proposed by Khadka *et al.* [25], the recommended range for infit and outfit mean square values was (0.7, 1.3).

3. Dimensionality: the scale is deemed unidimensional when there is a single latent variable of interest and the measurement focuses on the level of this variable [26]. To assess the dimensionality of the CVSS17$_{ENG}$, a Principal Component Analysis (PCA) of the residuals was conducted in WINSTEPS to calculate the raw variance explained by measures and the eigenvalue of the first contrast. Cases where less than 50% of the raw variance is explained by measures and/or the Eigenvalue of the first contrast is higher than two are considered indicative of potential multidimensionality [25,27]. If this occurs, further analyses are required. Flor this study, we decided to calculate the disattenuated correlation coefficient between the item clusters obtained from the PCA analysis [28]; according to Kim *et al.* [29], values above 0.8 indicates no benefit in considering the test as multidimensional.

4. Person separation index (PSI): The Rasch-based PSI, a reliability index -analogous to Cronbach's α used in Classical Test Theory- ranging from 0 to 1, implies acceptable reliability when its value exceeds 0.8 [30].

5. Levels of performance: the number of different levels of performance was calculated according to the method described by Wright [31].

6. Targeting: The alignment between the difficulty of the items and the participants' visual abilities was determined by calculating the difference between the average item difficulty and the average symptom level of the subjects [25,27].

7. Differential item functioning (DIF): We examined each item to check if there was any difference in the way subgroups (male–female; respondents aged < 40 years vs ≥ 40 years old) answered each item (i.e., no DIF). For this purpose, we conducted a DIF analysis using WINSTEPS, which is based on two methods [32]:

 a. The Mantel-Haenszel method, which estimates the log odds of DIF size and significance from cross-tabulations of observations in the two groups.

 b. The logit-difference (logistic regression) method, which estimates the difference in Rasch item difficulties between the two groups, holding all other factors constant.

According to Khadka *et al.* [33], DIF contrast (i.e., the difference in item difficulty between the two groups) was classified as no-DIF for contrasts less than 0.50 logits, minimal for contrasts between 0.50 and 1.0 logits, and notable for contrasts greater than 1.0 logits.

Moreover, because testing for DIF is a useful way to validate questionnaire translations [34], we used DIF analysis to test whether the CVSS17$_{ENG}$ items were equivalent to the original. DIF for an item was considered a cross-cultural or translational-related issue for that particular translation [35].

For readers unfamiliar with these variables, a more detailed description of each of the assessed variables is provided elsewhere [25,27].

The CVSS17 has a distinctive feature in that it provides a reliable overall score for the set of DES symptoms, while also allows the independent assessment of either the visual or the ocular symptoms attributable to computer use [4]. These subscales are called the *Internal Symptom Factor* (ISF), which comprises items A2, A22, A28, A30, A33, C21 and C24, and the *External Symptom Factor* (ESF), which encompasses items A4, A9, A17, A20, A21, A32, A33, B7, B8, C16 and C23.

To confirm, in the CVSS17$_{ENG}$, the domain structure proposed for the CVSS17, a PCA of the residuals was conducted for each subscale to confirm its unidimensionality.

## Results

### English version of CVSS17 (CVSS17$_{ENG}$)

The final version of the questionnaire arising from the pretest is available as supporting information (S2 File), along with its scoring table (S1 Table).

**Pretesting.** Participants for this part of the study were 61 subjects (age, 37.62 ± 12.52 years; range, 23–64 years; females, 57.4%; presbyopes: 41.0%) who completed the CVSS17$_{ENG}$. Item statistics for this preliminary Rasch analysis and the most relevant topics discussed in the Final Review Meeting are shown as supporting information (S1 File).

**Association between CVSS17$_{ENG}$ and VDS.** Table 1 summarizes the main results of CVSS17$_{ENG}$ and VDS in the pretest.

K-S test for CVSS17$_{ENG}$ and VDS scores (expressed in logits) indicated a normal distribution of both sets of measures (P > 0.05). Accordingly, we calculated the Pearson correlation

**Table 1. Summary of CVSS17$_{ENG}$ and Visual Discomfort Scale (VDS) results in the pretest sample, expressed as raw scores and logits.**

|  |  | CVSS17$_{ENG}$ raw score | CVSS17$_{ENG}$ (logits) | VDS raw score | VDS (logits) |
|---|---|---|---|---|---|
| **Responses** | valid | 61 | 61 |  | 56 |
|  | missing | 0 | 0 | 5 | 5 |
| **Mean** |  | 29.59 | -1.34 | 36.02 | -1.70 |
| **Median** |  | 30 | -0.96 | 33 | -1.88 |
| **Standard Deviation** |  | 6.50 | 1.83 | 8.43 | 1.34 |
| **Minimum** |  | 17 | -7.01 | 24 | -4.83 |
| **Maximum** |  | 47 | 2.57 | 56 | 0.64 |
| **Interquartile range** |  | 25 to 34 | -2.20 to 0.15 | 30.25 to 42 | -2.35 to -0.67 |

coefficient between total scores for CVSS17 and VDS at 0.54 (p < 0.01) (Fig 2), this value can be considered as evidence of convergent validity[25].

## Stage 2 results

For Stage 2, we utilized the responses from 441 questionnaires. To exclude the responses of subjects who did not answer the questionnaire sincerely (e.g., providing random answers), as we did in the development of CVSS17, we eliminated questionnaires with an Outfit > 2.5. Consequently, the responses of 19 participants were excluded from our dataset.

Finally, 422 questionnaires (50.24% female, 39.10% presbyopes, mean age ± SD was 38.57 ± 11.17, interquartile range for age was 30 to 46) were used in the analysis. The mean ± standard deviation CVSS17$_{ENG}$ score was 28.37 ± 7.22 and interquartile range was (23, 33).

**Item fit statistics.**  Item fit statistics, item measure (difficulty, in logits) and point bi-serial correlation for CVSS17$_{ENG}$ are displayed in Table 2.

**Dimensionality.**  Principal component analysis (PCA) of CVSS17$_{ENG}$ scores revealed that 49.8% of the raw variance was explained by CVSS17$_{ENG}$ measures. The first contrast eigenvalue was 2.19. To rule out multidimensionality [25,27], we examined the disattenuated correlation coefficient between the first and second contrast, which was 0.82. This indicates a 67% shared variance in person measures, so for practical purposes, both contrasts were considered as different branches of the same measures [29,36] and CVSS17$_{ENG}$ can be considered unidimensional [37].

**Person separation index and performance levels.**  The Person Separation Index for CVSS17$_{ENG}$ was 2.73, indicating a reliability of 0.88, distinguishing 3.97 score strata. Using the Wright method, a sample-independent method [31], CVSS17$_{ENG}$ was able to distinguish 5.7 symptom levels.

**Targeting.**  The targeting value estimated was -1.43 logits. The item-person map (Fig 3) shows that the CVSS17$_{ENG}$ were very demanding for the ability level in this sample, as we assessed a population-based sample, which included many individuals expected to have a low level of symptoms.

**Differential item functioning (DIF) by sex and age.**  Minimal DIF (DIF contrast = 0.50 for women) was found for item A21 ("Did your eyes burn?"). No more items showed either

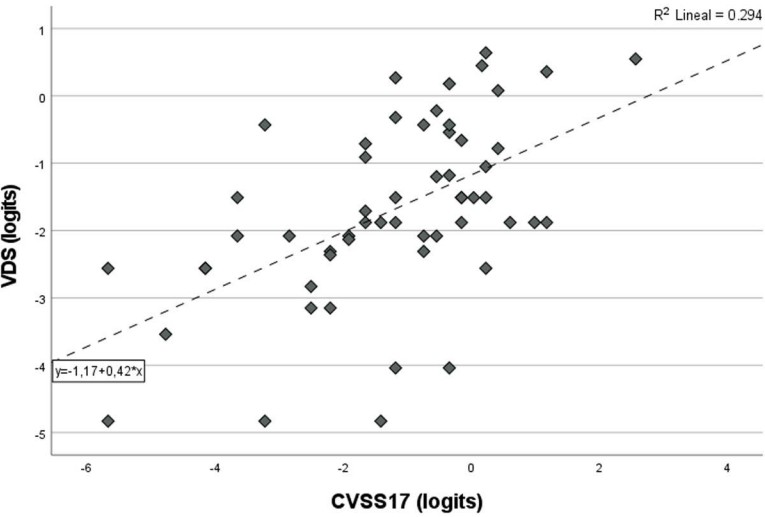

**Fig 2.  Scatter plot of correlation between CVSS17$_{ENG}$ and VDS.** Values are expressed in logits.

**Table 2. CVSS17$_{ENG}$ item fit statistics.**

| Item | Measure | Infit MnSQ | Outfit MnSQ | Pt. Bis |
|------|---------|------------|-------------|---------|
| A2 | 0.57 | 0.97 | 0.93 | 0.61 |
| A4 | -2.25 | 0.79 | 0.77 | 0.77 |
| A9 | 0.21 | 0.79 | 0.78 | 0.76 |
| A17 | -0.41 | 0.97 | 0.96 | 0.72 |
| A20 | -0.12 | 1.03 | 1.03 | 0.70 |
| A21 | -0.87 | 0.99 | 1.00 | 0.68 |
| A22 | 0.57 | 0.93 | 0.86 | 0.64 |
| A28 | 0.71 | 1.25 | 1.13 | 0.52 |
| A30 | 2.22 | 0.83 | 0.32 | 0.40 |
| A32 | 0.60 | 1.06 | 1.08 | 0.65 |
| A33 | -1.6 | 1.06 | 1.04 | 0.66 |
| B7 | -0.91 | 1.05 | 1.06 | 0.55 |
| B8 | 1.14 | 0.96 | 0.82 | 0.58 |
| C16 | -0.68 | 1.13 | 1.17 | 0.63 |
| C21 | 0.32 | 0.96 | 1.01 | 0.63 |
| C23 | 0.34 | 1.01 | 0.96 | 0.62 |
| C24 | 0.15 | 1.22 | 1.20 | 0.57 |

Results are ordered by item Id (left column). Measure = item difficulty (in logits); MnSQ = mean Square; Pt. Bis = point bi-serial correlation.

All items fell within the interval (0.7, 1.3) recommended by other authors[25,27] with the exception of A30, with an Outfit of 0.32.

minimal or notable DIF for sex, so we did not investigate this further as, according to Khadka *et al.* [25], one item with minimal DIF does not indicate a substantial reduction in the quality of a questionnaire.

According to age group (presbyopes vs. non presbyopes), two items showed minimal DIF: Item A30 ("Did the letters appear double?"), with DIF contrast=0.86 for presbyopes, and Item B7 ("Watery eyes"), with DIF = 0.56 for non-presbyopes. As indicated by Khadka et al. [25], the identification of minimal DIF in two items does not imply a substantial deterioration in the questionnaire's quality.

English version vs. Spanish version (based on DIF analysis and $R^2$ calculation)

We compared the responses of the second stage participants with the data from a sample of Spanish people used in a prior study [4]. The main characteristics of all the samples used in the study are summarized in Table 3.

DIF analysis showed that four items showed minimal DIF: In item A17, DIF contrast was 0.53 for the English version; in item A28, DIF contrast was 0.90 for the Spanish version; in A32, English version had DIF contrast of 0.75; in C24, Spanish version had 0.64 for the Spanish version. According to the mentioned guidelines [25], four items with minimal DIF in CVSS17$_{ENG}$ are acceptable, making both versions equivalent. In addition, the coefficient of determination ($R^2$) between the item measures derived from the DIF analysis was 0.856, so both versions shared 85.6% of their variability.

**Psychometric properties of CVSS17$_{ENG}$.** The main properties of CVSS17$_{ENG}$ compared with Rash model expectations are shown in Table 4.

According to existing guidelines [25,27], these results suggest that overall quality of CVSS17$_{ENG}$ is good.

**Convergent validity and repeatability.** CVSS17$_{ENG}$ – Ocular Comfort Index Spearman Rho correlation index was calculated at 0.66 (p < 0.001) (Fig 4).

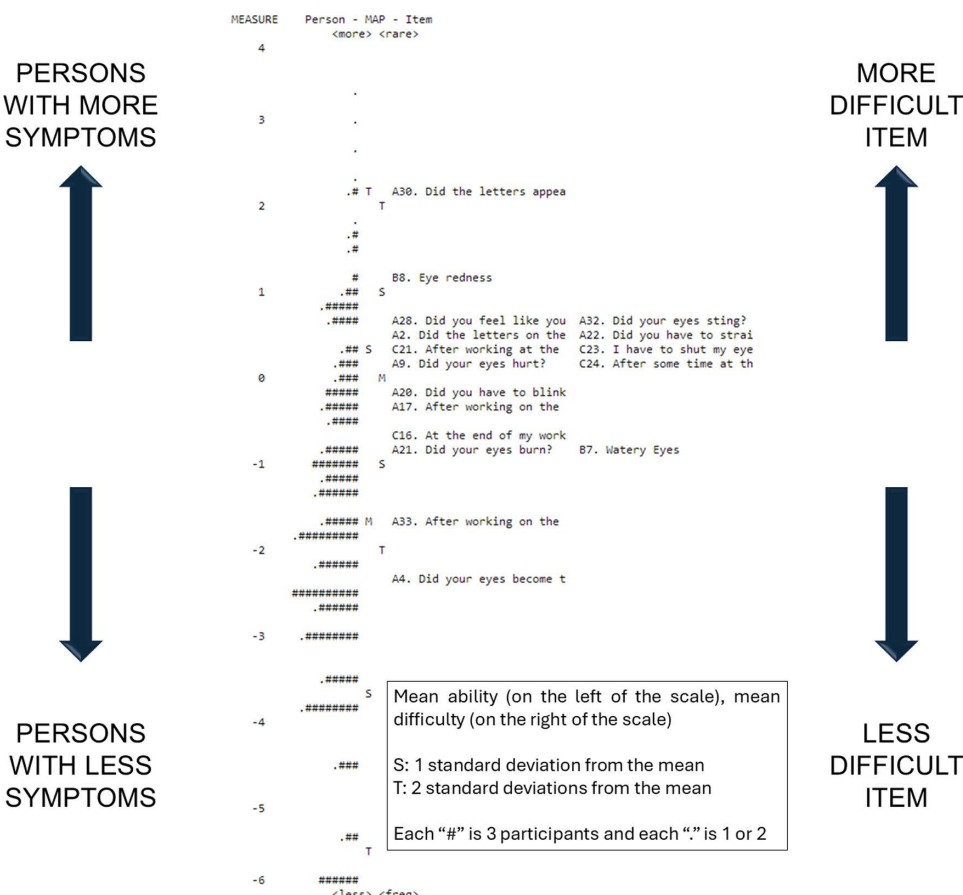

**Fig 3. Rasch item-person map, displaying the self-reported symptoms level of the patients in our study (left side) along with the corresponding item difficulty (right side).**

**Table 3. Main characteristics of the samples used in the study.**

| | | CVSS17 | CVSS17$_{ENG}$ | CVSS17$_{ENG}$ | CVSS17$_{ENG}$ |
| --- | --- | --- | --- | --- | --- |
| | | | Pre-test sample | Validation sample | Test-retest sample |
| **n** | | 796 | 61 | 422 | 218 |
| **Age** | **mean** | 43.93 | 37.62 | 38.57 | 42.06 |
| | **S.D.** | 10.05 | 12.52 | 11.17 | 11.78 |
| | **Interquartile range** | 36 to 52 | 27 to 50 | 30 to 46 | 33 to 52 |
| **Presbyopes (%)** | | 64.45 | 40.98 | 39.10 | 54.63 |
| **Female (%)** | | 58.04 | 57.38 | 50.24 | 46.51 |
| **Location** | | Spain | USA | USA | USA |
| **Mother tongue** | | Spanish | English | English | English |

The correlation coefficients values obtained can be considered as evidence of convergent validity [25,27], the overall results of the convergent validity assessment are summarized in Table 5.

For the test–retest assessment: the time interval between both administrations was 14.03 ± 0.11 days, the two-way single measure Intraclass Correlation Coefficient for test–retest

**Table 4. Comparison among CVSS17$_{ENG}$, the original CVSS17 and Rasch model expectations.**

| Parameter | Rasch model expectation | CVSS17$_{ENG}$ | CVSS17 |
|---|---|---|---|
| Numer of items | - | 17 | 17 |
| Response categories ordering | Ordered | Ordered | Ordered |
| Person separation index (reliability) | > 2.0 (> 0.80) | 2.73 (0.88) | 2.85 (0.89) |
| Item separation | > 3.0 | 8.91 | 8.61 |
| PCA Analysis: Eigenvalue of the first contrast | < 2.0 | 2.19 | 1.37 |
| Number of items with Infit outside (0.7, 1.3) | 0 | 0 | 0 |
| Number of items with Outfit outside (0.7, 1.3) | 0 | 0 | 0 |
| Number of items with DIF for gender of >1.0 logits and p < 0.05 | None | None | None |
| Number of items with DIF for age group of >1.0 logits and p < 0.05 | None | None | None |
| Targeting | < 1.0 | 1.43 | 0.89 |
| Intraclass Correlation Coefficient (ICC) | - | 0.79 (0.74, 0.84) | 0.85 (0.80, 0.89) |
| Coefficient of repeatability (from Bland-Altman plot) | - | 9.00 | 8.14 |

DIF = differential item functioning; PCA = principal component analysis of the residuals.

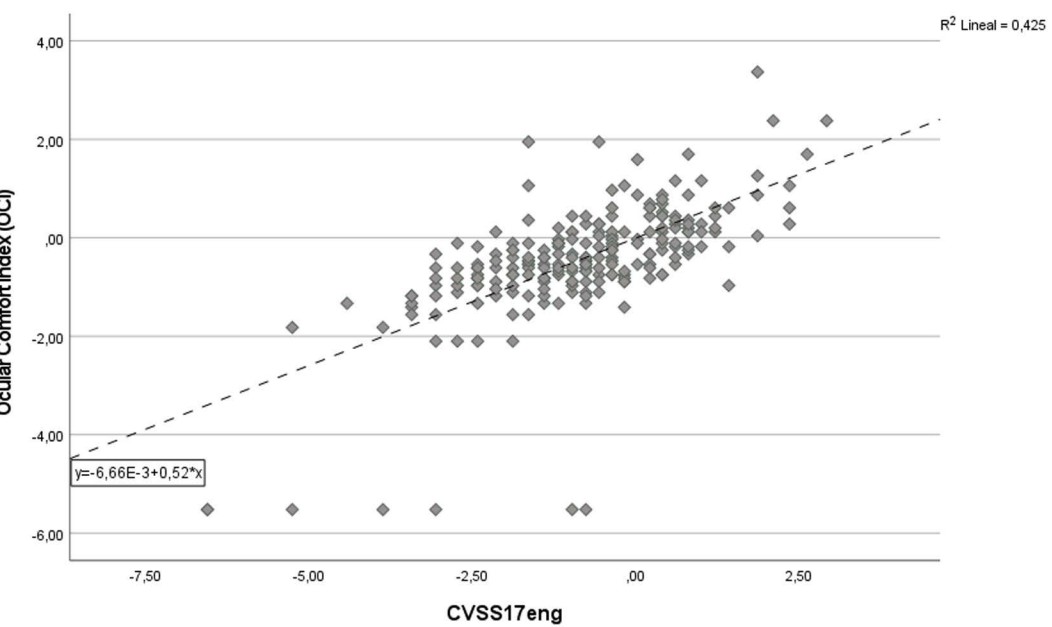

**Fig 4. Scatter plot of the association between CVSS17$_{ENG}$ and Ocular Comfort Index (OCI).** OCI and CVSS17$_{ENG}$ scores are expressed in logits.

repeatability was 0.794 (95% CI, 0.737–0.840), and within-subject standard deviations was 7.15. The mean difference between sessions was 0.39 and the limits of agreement including 95% of the differences were 9.39, -8.61 (Fig 5).

**Domain structure assessment.** For ISF, raw variance explained by measures was 51% and the Eigenvalue for the first contrast was 1.95, so ISF can be considered as unidimensional.

For ESF, the raw variance explained by measures was 53% and the Eigenvalue for the first contrast was 2.21. To rule out multidimensionality, we examined the disattenuated correlation

**Table 5. Summary of convergent validity assessment results.**

| | CVSS17$_{ENG}$ *vs.* VDS | CVSS17$_{ENG}$ *vs.* OCI |
|---|---|---|
| **Sample location** | USA | USA |
| **Test used for comparison** | Visual Discomfort Scale (VDS) | Ocular Comfort Index (OCI) |
| **Domain assesed** | Visual symptoms | Ocular Symptoms |
| **Coefficient of correlation** | Pearson | Spearman Rho |
| **Coefficient value (r)** | 0.54 | 0.66 |
| **95% confidence interval for r** | (0.34, 0.69) | (0.58, 0.73) |

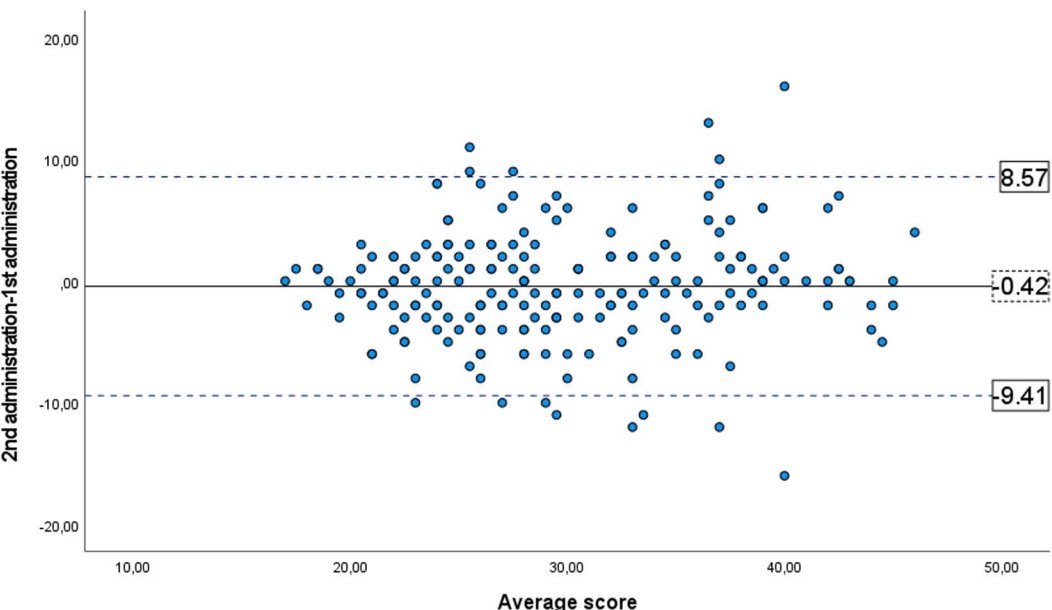

**Fig 5. Bland-Altman plot for CVSS17$_{ENG}$.** where the dotted line represents the mean difference between scores obtained from completing the questionnaire on two occasions. The solid lines represent the lower and upper 95% limits of agreement. The scores are expressed in raw score units.

coefficient between the first and second contrast, which was 0.78, indicating a 61% shared variance in person measures. Therefore, for practical purposes, both contrasts were considered as different branches of the same measures [36] and the subscale was judged unidimensional [37].

## Discussion

CVSS17$_{ENG}$ was developed through a standard valid process to ensure its equivalence with CVSS17 [17]. This will enable cross-cultural research in DES and will allow comparability of data obtained from different communities. Optometrists and Ophthalmologists can now use this valid and reliable instrument, fostering a more consistent assessment of DES symptoms even across language barriers.

The higher accuracy of Rasch-validated instruments [38] helps clinicians make informed decisions about patient care, based on robust and comparable data. Moreover, CVSS17$_{ENG}$ addresses a critical gap in existing PRO-instruments for assessing DES in English speaking populations, as it can independently measure the two dimensions of DES: ocular symptoms

and visual symptoms. This could enhance the precision and efficacy of therapeutic strategies for DES, because it allows tailoring interventions to target the unique aspects of visual discomfort experienced by individuals. In addition, epidemiological studies on DES may now use CVSS17$_{ENG}$ to analyze differences in ocular and visual symptoms among different populations and study their association with any possible risk factor.

It is important to note that DES should not be considered a disease or physiological condition. Instead, it is a set of symptoms that arise from prolonged use of screen-based digital devices. Unlike diseases, DES does not follow a binary presence/absence paradigm, but manifests along a spectrum of severity. Recognizing this subtle distinction is essential for accurate assessment and effective management. To determine the number of statistically significant different levels of scores (in our case, symptom scores) that a scale can discriminate, PCM analysis provides a reliability indicator, the Person Separation Index (PSI), which can be used to calculate the theoretical number of levels discriminated by the scale. Based on this data, the CVSS17$_{ENG}$ should be able to distinguish 3.97 levels [39]. However, this method may underestimate the effectiveness of the test if the person distribution is skewed towards the more "symptomatic side" of the scale [40], a common finding in PRO instruments for measuring symptoms. In such cases, the Wright method [40] helps to determine the true performance of the test. Our results showed that the CVSS17$_{ENG}$ can measure five different levels of performance. Unlike other existing questionnaires that categorize DES scores into dichotomous groups, the CVSS17$_{ENG}$ defines five Rasch-derived levels of symptoms, allowing for more detailed analysis that recognizes and quantifies subtle variations in symptom severity. This unique feature enables the differentiation and subsequent statistical study of different levels of DES symptoms, improving diagnostic accuracy and providing a framework for studying the progression of DES in clinical settings. In addition, researchers will gain a better understanding of the impact of DES on individuals in different context and demographics, which would be particularly useful for epidemiological studies.

During the pre-test, we received no negative feedback regarding the comprehensibility of any item, and there were just two missing responses in one item.

The assessment of convergent validity is essential when investigating the validity of a new PRO instrument, as it involves comparing the measures of a new test with those of another test that measures a closely related concept. As the DES is a group of symptoms related to ocular-surface complaints (referred to as "ocular symptoms") and another set of symptoms related to visual comfort (referred to as "visual symptom"), we assessed the convergent validity of CVSS17$_{ENG}$ by comparing its responses with those of a questionnaire measuring visual discomfort, the VDS [41], and of another one designed to assess dry eye, the OCI [42]. The coefficient of correlation between VDS and CVSS17 [3], 0.54, was close to the one calculated for VDS versus CVSS17$_{ENG}$, 0.60. Correlation between the Ocular Surface Disease Index (OSDI) and CVSS17 [43], 0.65, was almost equal to the one obtained for OSDI versus CVSS17$_{ENG}$ (0.66). These findings offer evidence of the convergent validity of CVSS17$_{ENG}$ and showed that the shared variance of CVSS17$_{ENG}$ with other scales measuring related topics is similar to those observed for CVSS17.

As recommended by Bradley and Massof [44], we used PCM results to compare item psychometric properties between CVSS17$_{ENG}$ and the original CVSS17. The results of this comparison confirmed the similarity between the two versions. Furthermore, we conducted a Rasch-based DIF analysis to compare the items of the Spanish version with those of the English version. This comparison is crucial for ensuring validity, as the absence of DIF indicates that both versions measure the same items is the same way. The presence of DIF in one item would imply that the measures obtained from the symptom assessed by this item is not equivalent in both versions. Additionally, the between-version DIF analysis is especially

helpful in the translation review, as the presence of DIF in any item may indicate an inaccurate translation. Then, the lack of significant DIF between the English and Spanish versions guarantees the validity of the CVSS17$_{ENG}$ and the quality of its translation.

The test-retest repeatability of CVSS17$_{ENG}$ was comparable to that of the Spanish version. Those seeking to maximize its accuracy, may achieve this by administering the questionnaire on two separate occasions instead of once.

CVSS17$_{ENG}$ fitted the model (Infit: 1.00, Outfit: 0.95) but Outfit value for item A30 ("Did the letters appear double?") was 0.32, suggesting limited measurement value and redundant information [45, 46]. However, no other items of CVSS17$_{ENG}$ showed misfitting so this can be considered acceptable [25,27]. Therefore, item A30 remained unchanged in the final version of CVSS17$_{ENG}$.

Principal Component Analysis returned an eigenvalue higher than two, which was not observed during CVSS17 development. This could be because we used BIGSTEPS instead of WINSTEPS and it also could be explained by differences in sample composition. Nevertheless, the unidimensionality of CVSS17$_{ENG}$ was confirmed, for statistical purposes, through the disattenuated correlation coefficient between the first and second contrast from Principal Component Analysis.

One limitation of the study is the lack of clinical information from the participants, such as whether participants wore glasses or had any eye diseases. As a result, it was not possible to evaluate the impact of this clinical data on the CVSS17$_{ENG}$ performance.

## Conclusions

The adaptation of the CVSS17 into English (CVSS17$_{ENG}$) was rigorously conducted, ensuring linguistic equivalence and psychometric robustness.

Rasch analysis confirmed that CVSS17$_{ENG}$ is reliable, unidimensional, and performs equivalently to the original Spanish version.

Minimal DIF suggests cross-cultural equivalence and convergent validity was established through correlations with the VDS and the OSDI.

The study confirms that CVSS17$_{ENG}$ is a reliable tool for assessing DES in English-speaking populations in both research and clinical settings.

## Supporting information

**S1 File. Expert committee report, describing the final review meeting for the cross-cultural adaptation.** The name and role of each participant are provided, along with a summary of the discrepancies discussed in the session. Summary statistics for the Rasch analysis conducted on the pre-test responses are also displayed.
(PDF)

**S2 File. Hard copy version of the English version of Computer Vision Symptom Scale (CVSS17$_{ENG}$).** PDF version of the CVSS17$_{ENG}$, for those interested in distributing it as a hard copy.
(PDF)

**S1 Table. Scoring chart for the CVSS17ENG.** Each CVSS17$_{ENG}$ item is identified by a capital letter and a one- or two-digit number. Additionally, each response option for each item is identified by a number. This table is necessary to determine the score that each subject gets for each question when using the hard-copy version (respondents are automatically scored in the online version). For example, if a respondent chooses option 6 for the first question, they will receive three points; choosing option 3 for item A30 gets one point. The final score for each

subject is obtained using the formula at the bottom of the table, both in CVSS17 points and in logits.
(PDF)

## Acknowledgements

The authors thank the SUNY's staff for their cooperation in the first stage if the study.

## Author contributions

**Conceptualization:** Mariano González-Pérez, Carlos Pérez-Garmendia, Kathleen Hoang, Rosario Susi, Beatriz Antona, Ana-Rosa Barrio, Mark Rosenfield.

**Data curation:** Mariano González-Pérez, Carlos Pérez-Garmendia, Kathleen Hoang, Rosario Susi.

**Formal analysis:** Mariano González-Pérez, Carlos Pérez-Garmendia, Rosario Susi.

**Funding acquisition:** Mariano González-Pérez, Beatriz Antona, Ana-Rosa Barrio.

**Investigation:** Mariano González-Pérez, Kathleen Hoang.

**Methodology:** Mariano González-Pérez, Carlos Pérez-Garmendia, Rosario Susi, Beatriz Antona, Ana-Rosa Barrio, Mark Rosenfield.

**Project administration:** Beatriz Antona, Ana-Rosa Barrio.

**Resources:** Mariano González-Pérez, Carlos Pérez-Garmendia, Kathleen Hoang, Ana-Rosa Barrio.

**Software:** Mariano González-Pérez, Carlos Pérez-Garmendia.

**Supervision:** Mariano González-Pérez, Rosario Susi, Beatriz Antona, Mark Rosenfield.

**Validation:** Mariano González-Pérez, Rosario Susi, Beatriz Antona, Ana-Rosa Barrio, Mark Rosenfield.

**Visualization:** Mariano González-Pérez, Carlos Pérez-Garmendia, Kathleen Hoang, Rosario Susi, Beatriz Antona, Ana-Rosa Barrio, Mark Rosenfield.

**Writing – original draft:** Mariano González-Pérez.

**Writing – review & editing:** Mariano González-Pérez, Carlos Pérez-Garmendia, Kathleen Hoang, Rosario Susi, Beatriz Antona, Ana-Rosa Barrio, Mark Rosenfield.

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
