## [Decision Letter · Decision Letter 0]

30 Apr 2024

PONE-D-24-02434English version of the Computer Vision Symptom Scale (CVSS17): translation and Rasch analysis-based cultural adaptationPLOS ONE

Dear Dr. González-Pérez,

Thank you for submitting your manuscript to PLOS ONE. After careful consideration, we feel that it has merit but does not fully meet PLOS ONE’s publication criteria as it currently stands. Therefore, we invite you to submit a revised version of the manuscript that addresses the points raised during the review process.

We look forward to receiving your revised manuscript.

Kind regards,

Amir H. Pakpour, Ph.D.

Academic Editor

PLOS ONE

[This study was supported by the Instituto de Salud Carlos III through Project “PI18/00374” (co-funded by European Regional Development Fund “A way to make Europe”).]

 [BA, AB, MGP and RS received a grant by the Instituto de Salud Carlos III (https://www.isciii.es/Paginas/Inicio.aspx)  through Project “PI18/00374” (co-funded by European Regional Development Fund “A way to make

Europe”). The funders didn't play any role in the study design, data collection and analysis, decision to publish, or preparation of the manuscript]

3. Thank you for uploading your study's underlying data set. Unfortunately, the repository you have noted in your Data Availability statement does not qualify as an acceptable data repository according to PLOS's standards.

Reviewers' comments:

Reviewer's Responses to Questions

**Comments to the Author**

1. Is the manuscript technically sound, and do the data support the conclusions?

Reviewer #1: Partly

2. Has the statistical analysis been performed appropriately and rigorously? 

Reviewer #1: No

3. Have the authors made all data underlying the findings in their manuscript fully available?

Reviewer #1: Yes

4. Is the manuscript presented in an intelligible fashion and written in standard English?

Reviewer #1: Yes

5. Review Comments to the Author

Reviewer #1: comments to improve the paper.

provide more details on the translation and cross-cultural adaptation process. specifically, describe the steps taken to ensure cultural equivalence and appropriateness of the translated items.

clarify the inclusion and exclusion criteria for the participants in the two stages of the study.

provide more information on the recruitment process, especially for the online panel used in stage 2. Justify sample size for each.

statistical analyses need major rework for explanations. the rationale for choosing the partial credit model (pcm) over the rating scale model (rsm) for rasch analysis should be explained in more detail. I am confused if this is irt or mirt?

the method used to assess unidimensionality should be described more clearly. the authors mention using the disattenuated correlation coefficient, but it would be helpful to provide a brief explanation of this approach. Use updated refs.

the authors should consider providing more details on the assessment of differential item functioning (dif) and the interpretation of the dif contrast values. I am lost in present manuscript.

the interpretation of the person separation index (psi) and the levels of performance should be clarified for readers who may be less familiar with rasch analysis.

the authors should explain the criteria used to determine acceptable fit statistics (e.g., the recommended range for infit and outfit mean square values).

provide more information on the methods used to assess convergent validity, specifically the rationale for choosing the visual discomfort scale (vds) and the ocular comfort index (oci) as comparison measures.

the authors should consider presenting the correlation coefficients between cvss17eng and vds/oci with their corresponding confidence intervals.

results and discussion it would be helpful to include a table summarizing the demographic characteristics of the participants in both stages of the study.

the results for the convergent validity assessment with vds and oci could be presented more clearly, perhaps with additional figures or tables. better visually.

the authors should consider discussing the clinical significance or implications of the observed differences in dif between the english and spanish versions.

6. PLOS authors have the option to publish the peer review history of their article (what does this mean? ). If published, this will include your full peer review and any attached files.

**Do you want your identity to be public for this peer review?** For information about this choice, including consent withdrawal, please see our Privacy Policy .

Reviewer #1: No

---

## [Author Response · Author response to Decision Letter 1]

17 Jun 2024

Response to Journal requirements:

Please ensure that your manuscript meets PLOS ONE's style requirements, including those for file naming. The PLOS ONE style templates can be found at https://journals.plos.org/plosone/s/file?id=wjVg/PLOSOne_formatting_sample_main_body.pdf and https://journals.plos.org/plosone/s/file?id=ba62/PLOSOne_formatting_sample_title_authors_affiliations.pdf.

According to your advice, we revised file naming and addressed some minor formatting issues, such as changing parentheses to brackets in the references.

[This study was supported by the Instituto de Salud Carlos III through Project “PI18/00374” (co-funded by European

Regional Development Fund “A way to make Europe”).]

We note that you have provided funding information that is not currently declared in your Funding Statement.

However, funding information should not appear in the Acknowledgments section or other areas of your manuscript.

We will only publish funding information present in the Funding Statement section of the online submission form. Please remove any funding-related text from the manuscript and let us know how you would like to update your Funding Statement. Currently, your Funding Statement reads as follows:

[BA, AB, MGP and RS received a grant by the Instituto de Salud Carlos III (https://www.isciii.es/Paginas/Inicio.aspx) through Project “PI18/00374” (co-funded by European Regional Development Fund “A way to make

Europe”). The funders didn't play any role in the study design, data collection and analysis, decision to publish, or preparation of the manuscript]

Thank you for your advice. We have omitted this information from the acknowledgements section and are providing a new Funding Statement along with the cover letter. The new Funding Statement remains at follows: “This study was supported by a grant by the Instituto de Salud Carlos III (https://www.isciii.es/Paginas/Inicio.aspx) through Project “PI18/00374” (co-funded by the European Regional Development Fund “A way to make Europe”). The funders didn't play any role in the study design, data collection and analysis, decision to publish, or preparation of the manuscript”

3. Thank you for uploading your study's underlying data set. Unfortunately, the repository you have noted in your DataAvailability statement does not qualify as an acceptable data repository according to PLOS's standards.

Thank you. According to your advice, we’ve uploaded to FigShare the minimal dataset necessary to replicate our study findings. The new dataset’s DOI (10.6084/m9.figshare.25909171) is provided in the corresponding submission section.

4. Please include captions for your Supporting Information files at the end of your manuscript, and update any in-textcitations to match accordingly. Please see our Supporting Information guidelines for more information: http://journals.plos.org/plosone/s/supporting-information.

Thank you. We missed including it in the first version, so in the new version of the manuscript, a list of the Supporting Information captions is provided at the end of the manuscript in a section titled 'Supporting Information'

1. Response to review Comments to the Author

Reviewer #1: comments to improve the paper.

• provide more details on the translation and cross-cultural adaptation process. specifically, describe the steps taken to ensure cultural equivalence and appropriateness of the translated items.

Thank you for your suggestion. Based on it we have provided this information in lines 121-127

• clarify the inclusion and exclusion criteria for the participants in the two stages of the study.

According to your recommendation, we’ve clarified the inclusion criteria for stage 1 (lines 149-160) and for stage 2 (lines 177-187)

• provide more information on the recruitment process, especially for the online panel used in stage 2.

Thank you for your suggestion. Based on it we have provided more information about the online panel used in stage 2. (lines 169-174)

• Justify sample size for each

Thank you for your feedback. Regarding sample size justification:

The justification for sample size used in Stage 1 is described in lines 146-150

Justification for sample size used in the validation of the CVSS17ENG and in DIF analysis is described in lines 189-191

Justification for sample size used in the convergent validity and test-retest reliability assessment is displayed in lines 198-202.

statistical analyses need major rework for explanations.

• the rationale for choosing the partial credit model (pcm) over the rating scale model (rsm) for rasch analysis should be explained in more detail.

Thank you for your suggestion. Based on your feedback, we have provided a more detailed explanation (lines 230-241) of the rationale for choosing the Partial Credit Model (PCM) over the Rating Scale Model (RSM).

• I am confused if this is irt or mirt?

Thank you for your observation. In the revised manuscript (lines 225-228), we have clarified that we used a unidimensional Item Response Theory (IRT) model.

• the method used to assess unidimensionality should be described more clearly. the authors mention using the disattenuated correlation coefficient, but it would be helpful to provide a brief explanation of this approach. Use updated refs.

Thank you. Following your recommendation, we have added a brief explanation (lines 258-262) about the use of the disattenuated correlation coefficient for assessing unidimensionality when the PCA analysis suggests multidimensionality.

• the authors should consider providing more details on the assessment of differential item functioning (dif) and the interpretation of the dif contrast values. I am lost in present manuscript.

Thank you for your observation. Based on your feedback, we have included I the new manuscript more details abaut the DIF assessment and its interpretation (lines 273-283)

• the interpretation of the person separation index (psi) and the levels of performance should be clarified for readers who may be less familiar with rasch analysis.

According to your suggestion, we have included in the new manuscript a comment in line 264, and a brief section in the discussion (lines 437-455).

• the authors should explain the criteria used to determine acceptable fit statistics (e.g., the recommended range for infit and outfit mean square values).

Based on your feedback, the recommended range for infit and outfit mean square values are now described in the methods section of the manuscript (lines 248-250).

• provide more information on the methods used to assess convergent validity, specifically the rationale for choosing the visual discomfort scale (vds) and the ocular comfort index (oci) as comparison measures.

According to your observation, we discuss it in the new manuscript (482-489).

• the authors should consider presenting the correlation coefficients between cvss17eng and vds/oci with their corresponding confidence intervals.

Thank you for the suggestion. According to it, we have included a new table (Table 5) in the manuscript (lines 411-415) that summarizes the results of the convergent validity assessment, including the confidence intervals of the correlation coefficients.

results and discussion

• it would be helpful to include a table summarizing the demographic characteristics of the participants in both stages of the study.

Based on your feedback, we have modified Table 3 and know it provides the demographic characteristics of all the samples used in the study

• the results for the convergent validity assessment with vds and oci could be presented more clearly, perhaps with additional figures or tables. better visually.

According to your suggestion, we have included a new table (Table 5) in the manuscript (lines 411-415) that summarizes the results of the convergent validity assessment and completes the information displayed in Fig 2 and Fig 4. Thank you

• the authors should consider discussing the clinical significance or implications of the observed differences in dif between the english and spanish versions.

Thank you very much for your suggestions to improve the paper, according to your feedback we’ve added a paragraph in the discussion (lines 498-507) about the implications of the DIF analysis results.

---

## [Decision Letter · Decision Letter 1]

9 Jan 2025

English version of the Computer Vision Symptom Scale (CVSS17): translation and Rasch analysis-based cultural adaptation

PONE-D-24-02434R1

Dear Dr. González-Pérez,

We’re pleased to inform you that your manuscript has been judged scientifically suitable for publication and will be formally accepted for publication once it meets all outstanding technical requirements.

Kind regards,

Marianne Clemence

Staff Editor

PLOS ONE

Additional Editor Comments (optional):

Reviewers' comments:

Reviewer's Responses to Questions

**Comments to the Author**

1. If the authors have adequately addressed your comments raised in a previous round of review and you feel that this manuscript is now acceptable for publication, you may indicate that here to bypass the “Comments to the Author” section, enter your conflict of interest statement in the “Confidential to Editor” section, and submit your "Accept" recommendation.

Reviewer #1: All comments have been addressed

Reviewer #2: All comments have been addressed

2. Is the manuscript technically sound, and do the data support the conclusions?

Reviewer #1: Yes

Reviewer #2: Yes

3. Has the statistical analysis been performed appropriately and rigorously? 

Reviewer #1: Yes

Reviewer #2: Yes

4. Have the authors made all data underlying the findings in their manuscript fully available?

Reviewer #1: Yes

Reviewer #2: Yes

5. Is the manuscript presented in an intelligible fashion and written in standard English?

Reviewer #1: Yes

Reviewer #2: Yes

6. Review Comments to the Author

Reviewer #1: Thank you for addressing all raised concerns during the peer review.

Thank you for addressing all raised concerns during the peer review.

Reviewer #2: "The authors have demonstrated a thorough and thoughtful approach in addressing all the comments and concerns raised by the previous reviewer. They have carefully considered each point of feedback and have made significant revisions, clarifications, and improvements to the manuscript. These revisions include refining the structure and clarity of the writing, enhancing the methodological explanations, and providing more robust justifications for the choices made in their study.

The authors have also strengthened the theoretical framework by incorporating additional references and discussing their research in the context of the latest developments in the field. Furthermore, they have addressed concerns regarding the data analysis by providing clearer explanations, revisiting certain analytical steps, and including supplementary material to ensure transparency and rigor.

In response to concerns regarding the presentation of results, the authors have reorganized the relevant sections, added clearer visuals (e.g., figures, tables), and provided more detailed explanations to ensure that their findings are presented in a way that is both accessible and comprehensive for readers. These changes significantly improve the overall readability and scholarly quality of the paper.

Upon reviewing the updated manuscript, I am confident that the authors have adequately resolved all previous issues and that the paper now meets the necessary standards for publication. The improvements made reflect a high level of academic diligence, and the paper is now far stronger in terms of clarity, methodological rigor, and overall contribution to the field. Therefore, I am pleased to recommend that the paper be accepted for publication."

7. PLOS authors have the option to publish the peer review history of their article (what does this mean? ). If published, this will include your full peer review and any attached files.

**Do you want your identity to be public for this peer review?** For information about this choice, including consent withdrawal, please see our Privacy Policy .

Reviewer #1: No

Reviewer #2: **Yes: ** Dr. Ragni Kumari

---

## [Editor Report · Acceptance letter]

PONE-D-24-02434R1

PLOS ONE

Dear Dr. González-Pérez,

I'm pleased to inform you that your manuscript has been deemed suitable for publication in PLOS ONE. Congratulations! Your manuscript is now being handed over to our production team.

Kind regards,

on behalf of

Dr. Amir H. Pakpour

Academic Editor

PLOS ONE